# Parking Time Violation Tracking Using YOLOv8 and Tracking Algorithms

**DOI:** 10.3390/s23135843

**Published:** 2023-06-23

**Authors:** Nabin Sharma, Sushish Baral, May Phu Paing, Rathachai Chawuthai

**Affiliations:** 1Department of Robotics and AI, School of Engineering, King Mongkut’s Institute of Technology Ladkrabang, Bangkok 10520, Thailand; 64601008@kmitl.ac.th (N.S.); sushish.ba@kmitl.ac.th (S.B.); 2Department of Biomedical Engineering, School of Engineering, King Mongkut’s Institute of Technology Ladkrabang, Bangkok 10520, Thailand; may.pa@kmitl.ac.th; 3Department of Computer Engineering, School of Engineering, King Mongkut’s Institute of Technology Ladkrabang, Bangkok 10520, Thailand

**Keywords:** DeepSORT, OC-SORT, object detection, tracking algorithm, vehicle tracking, YOLOv8

## Abstract

The major problem in Thailand related to parking is time violation. Vehicles are not allowed to park for more than a specified amount of time. Implementation of closed-circuit television (CCTV) surveillance cameras along with human labor is the present remedy. However, this paper presents an approach that can introduce a low-cost time violation tracking system using CCTV, Deep Learning models, and object tracking algorithms. This approach is fairly new because of its appliance of the SOTA detection technique, object tracking approach, and time boundary implementations. YOLOv8, along with the DeepSORT/OC-SORT algorithm, is utilized for the detection and tracking that allows us to set a timer and track the time violation. Using the same apparatus along with Deep Learning models and algorithms has produced a better system with better performance. The performance of both tracking algorithms was well depicted in the results, obtaining MOTA scores of (1.0, 1.0, 0.96, 0.90) and (1, 0.76, 0.90, 0.83) in four different surveillance data for DeepSORT and OC-SORT, respectively.

## 1. Introduction

In Thailand, due to the growing number of vehicular registrations, parking has become a serious issue and has introduced an unsolved challenge [1]. The rapid increase in car registrations in Thailand has led to parking problems in cities throughout Thailand. In order to meet the need for parking spaces, shopping malls and mini-marts are trying to effectively manage the limited number of parking spaces that are accessible to the public. In Thailand, mini-marts such as Tesco Express, 7-Eleven, and Mini Big C face a substantial parking problem due to the limited availability of parking spaces. In a fast-paced environment such as 7-Eleven with limited parking facilities, it is essential to track the parking time of each vehicle in the parking lot. This issue of parking violations has attracted the attention of many mini-mart executives and the general public.

Parking violations include activities such as parking a vehicle in a restricted area and time limit violations. Figure 1 shows an example of the implementation of a time restriction parking rule in a mini-mart in Thailand. Most shopping malls and mini-marts use closed-circuit television (CCTV) in parking lots to observe parking time violations. Currently, parking violations are manually checked by the relevant authorities. However, these methods are expensive and incur high labor costs, due to the constant monitoring required to track incoming and outgoing vehicles. To address the parking space problem, various existing parking solutions have used IoT devices to provide drivers with real-time parking details [2]. However, using sensors and hardware, while providing accuracy, also requires constant maintenance, making it an unideal solution for mini-marts. A low cost parking violation detection system is still required.

The majority of currently available research focuses on assisting drivers to locate parking spots. Numerous models have been developed to assist drivers with nearby parking spaces in real time [3]. However, there are very few studies that have been conducted to help authorities to manage parking spots effectively. Very few studies have been conducted regarding parking violations and time restrictions.

Over the past few years, mini-marts in Thailand have adopted the method of allocating a parking time of 15 min per vehicle outside their stores, in order to keep up with parking demand. However, this method has not been effective, as additional manpower is required to record the arrival time and departure time of each car.

Recently, considerable effort has been made to use CCTV cameras to identify illegal parking practices. Many research studies have employed a Gaussian Mixture Model-based image segmentation approach to retrieve vehicle information [4,5]. Akhawaji et al. [4] further utilizing a Kalman filter to eliminate false alarms and enhance efficient vehicle tracking. However, the effectiveness of this strategy might be affected if the lighting conditions in the operating area change. Apart from image processing algorithms, many Deep Learning algorithms have been deployed to detect parking violations. For instance, networks such as the Single Shot MultiBox Detector (SSD) and You Only Look Once (YOLO) have been implemented to identify parking violations [6,7,8].

This paper introduces a real-time parking time violation tracking algorithm using closed-circuit cameras and DL models with a tracking algorithm to persist the information from one frame to its subsequent frame. The algorithm leverages the state-of-the-art object detection technique, YOLOv8, to identify vehicles within a parking lot. Each detected vehicle is then assigned a unique ID in the parking lot using DeepSORT/OC-SORT, enabling efficient monitoring of parking time violations. To ensure good performance across varying levels of illumination, weather conditions, and short-term obstructions, optimal parameters were carefully selected for the algorithms. An important advantage of this algorithm is that it does not require prior knowledge about the Region of Interest (ROI), making it adaptable to various parking lot scenarios. This research paper presents the implementation of parking violation detection in a time-series context. The paper’s major contributions can be summarized as follows:

(1) The proposed algorithm demonstrates effective vehicle detection and tracking across various natural conditions, exploring the SOTA object detection and tracking models;

(2) The methodology exhibits an accurate and reliable solution to the parking time violation problem;

(3) This paper introduces a frame-by-frame evaluation with respect to time, which is a new approach to the problem domain.

The remaining sections of this paper are organized as follows. Section 2 reviews the background and literature related to object detection and object tracking. Section 3 describes the details of the proposed parking time violation process, including the experimental setup and experimental settings of parameters for the algorithms. Section 4 presents the results and discussion.

## 2. Related Work

Deep Learning applications have received significant traction due to their extensive research and development. Deep Learning-based methods have been applied in many tasks, such as classification [9], object detection [10], object tracking [11], and healthcare [12,13,14]. The objective of parking time violation tracking is to identify parking time violations by vehicles, especially in mini-marts which have a limited parking time. The task involves implementation of object detection and object tracking. Therefore, the current literature on object detection and object tracking is discussed accordingly.

### 2.1. Object Detection

In its early years, the machine learning-based object detection pipeline consisted of two primary steps: ROI extraction and object classification [15]. Based on Zhao et al. [16], object detection models consist of three phases: selection of region of interest, extraction of features, and, finally, classification of objects. One common ROI extraction technique uses sliding windows that move various scales across an image. Exhaustive sliding is, however, very challenging to apply in practice due to its great processing complexity.

Deep Learning algorithms are extensively implemented for object detection tasks. They can mainly be categorized into two types, as shown in Figure 2. You Only Look Once (YOLO) [17] and Single Shot MultiBox Detector (SSD) [18] consider detection tasks as a regression problem and are one stage networks. On the other hand, algorithms such as Region-Based CNN (RCNN) [10] firstly locate the region of interest, which is further classified into classes. As depicted in Figure 3, The R-CNN model uses a selective search algorithm to determine the number of candidates for bounding box object regions, and then features are fed into CNN, which acts as a feature extractor. Support Vector Machine (SVM) is used to classify whether the object is present within that candidate region by utilizing the retrieved features. R-CNN can perform well in various object detection tasks, but training these models takes significant time, and the speed of detection is limited [19].

To improve the speed of training and inference, one-stage detectors such as SSD and YOLO have been introduced for object detection tasks. Figure 4 shows the architecture of SSD Models. In SSD, CNN-based feature extractors are used. At the end of feature extraction, convolutional feature layers generate predictions at multiple scales. Since SSD does not use region based proposals, SSD models have increased detection speed, as opposed to two-stage detectors such as R-CNN. In this research work, YOLO was implemented for the object detection task. The details of the YOLO model are discussed accordingly in the Methodology section.

One of the first attempts to tackle the issue of parking lot monitoring using machine learning utilized color vector features on an SVM classifier to distinguish parking spaces inside a parking lot in 2002 [20]. Based on a 3D model of parking spaces, Huang et al. [21] proposed a day and night operating parking space detection system using a Bayesian hierarchical framework. Apart from Deep Learning models, many image processing techniques have been implemented to solve parking space detection problems. For instance, Menéndez et al. [22] proposed temporal analysis of parking area video frames to identify vacant spaces. This method includes the process of background subtraction using the Gaussian mixture to identify and track vehicles in the parking lot. In addition, a transience map was created to observe incoming and outgoing vehicles. Xie et al. [7] proposed an optimized SSD to detect illegal vehicle parking from a video stream in a robust environment, achieving 99% accuracy.

Recently, Patel and Meduri [23] presented an automatic parking space detection algorithm that comprised two steps. The first step included vehicle detection with Faster R-CNN and YOLOv4, while vehicle tracking was used to differentiate between moving and stationary vehicles. Based on Patel and Meduri’s research [23], this technique reduced the amount of human effort required by up to 90%. Grbić and Koch [24] proposed occupancy classification and a parking space detection algorithm, wherein parking spaces were determined as occupied or empty using a trained ResNet34 deep classifier; they extensively evaluated this approach on publicly known parking datasets such as PKLot [25] and CNRPark+EXT [26]. Chen et al. [27] implemented an enhanced SSD (Single Shot MultiBox Detector) algorithm for the quick recognition of vehicles in traffic scenarios. For feature extraction, the authors used the MobileNet v2 network, which helped to improve the detection accuracy of the algorithm. Experimental results showed that the proposed algorithm achieved 82.59% and 84.83% accuracy for the BDD100K and KITTI datasets, respectively. By introducing asymmetric convolution and a global point tracking module, Li et al. [28] proposed YOLO–GCC based on the YOLO algorithm and achieved an average accuracy of 83.0% on the TSD–MAX dataset. Jung et al. [29] proposed classification and localization of vehicles in traffic monitoring, using ResNet50 for vehicle classification, adding an optimized drop-down convolutional neural network (DropCNN) to improve the performance of classification.

### 2.2. Vehicle Tracking

Object tracking is a technique for detecting objects across frames by utilizing their spatial and temporal characteristics. In its simplest form, the method consists of obtaining the initial set of detections, giving them distinct IDs, and following them over frames, which is the essence of object tracking. Over the last decade, object tracking methods have gained popularity in the fields of Deep Learning and computer vision.

Single Object Tracking (SOT) and Multiple Object Tracking (MOT) are the two categories into which object tracking can be subdivided [30]. A Multiple Object Tracking algorithm’s primary responsibility is to identify multiple objects in a frame, assign and preserve their identities, and follow the object’s trajectory across input frames.

Object tracking has been used in many domains such as pedestrian tracking [31], vehicle tracking [32], and player tracking [33]. Parico and Ahamed [34] implemented YOLOv4 for pear detection and a multiple object tracking algorithm, DeepSORT, for pear tracking and counting. Hou et al. [32] proposed a tracking algorithm called DeepSORT and a low-confidence search filter that reduced false detections produced by the original DeepSORT algorithm. To track vehicles detected by a YOLOv3 network, Liu and Liu [35] proposed 3-D constrained multiple kernels aided by Kalman filtering. In this article, we focus on a closed-circuit television (CCTV)-based multiple object tracking technique, which allows identification of multiple objects in a sequence of images. The effectiveness of such a tracking method depends on the quality of detection in various weather settings, the proportion of occlusion, and illumination change. Therefore, it is crucial to choose the best object detection and tracking algorithm. Aware of our objective, state-of-the-art tracking algorithms such as DeepSORT and OCSORT were implemented. Table 1 shows the benchmark results on the DanceTrack dataset. OC-SORT outperforms all the existing tracking algorithm in many metrics, such as HOTA, AssA and IDF1. This result shows strong evidence of the high performance of the OC-SORT Algorithm.

## 3. Proposed Parking Time Violation Algorithm

The proposed algorithm utilizes a state-of-the-art object detection algorithm called YOLOv8 [39] for detecting vehicles. For tracking vehicles, two object tracking algorithms were implemented, Deep SORT (Simple Online Real Tracking) [11] and Observation-Centric SORT (OC-SORT) [38]. In-depth discussion of the algorithm is included in the subsection below.

### 3.1. Dataset

Due to limitations and privacy concerns, acquiring mini-mart data was not possible. As such, the dataset was comprised of video footage obtained from CCTV cameras installed at King Mongkut’s Institute of Technology, Ladkrabang (KMITL). The CCTV model utilized at KMITL is the Panasonic V series, which offers a 15–30 frames per second (FPS) capture rate and a 1080P resolution. These cameras are IP66 rated, providing resistance to water and dust.

Data collection was conducted using 4 cameras placed at different locations within the university campus, all with a resolution of 1080P and a frame rate of 15 FPS. The selection of dataset locations was carefully carried out to incorporate diverse camera angles and viewpoints. The video samples within the dataset encompassed various daylight and weather settings, offering a comprehensive representation of the parking lot scenarios at KMITL.

### 3.2. Vehicle Detection

In 2015, Joseph Redmon and Ali Farhadi from the University of Washington created the cutting-edge object recognition algorithm called YOLO (You Only Look Once) [17]. YOLO outperforms many other object detection algorithms such as R-CNN and DPM [17]. YOLO sees the entire image during training and testing, in contrast to sliding window- and region proposal-based approaches, implicitly capturing contextual information about classes as well as their appearance. The initial version of YOLO had the ability to quickly recognize objects in images, but it had trouble locating smaller objects precisely. Since the data in our problem domain does not contain very small images, there is no problem using YOLO for this study. YOLO divides input images into a grid of dimensions M × M. For instance, a grid cell is in charge of detecting an object if its center falls inside that grid cell. Each grid cell predicts the bounding box and offers a confidence score for the associated boxes. In YOLO, confidence is described as *Pr(Object) × IOU* where *Pr(Object)* represents the probability of the presence of an object, and IOU represents the Intersection over Union (IOU), i.e., the overlap area between inference and ground truth. Each grid cell generates five predictions (*x, y, w, h*, and a confidence score). Additionally, each grid produces p conditional class probabilities, expressed as *Pr(Class|Object)*. Equation (Equation 2) below demonstrates the method to obtain class-specific confidence scores for each box during the test phase.
(1)$$\begin{array}{c} {\mathrm{Class}}_{i}\times {\mathrm{IOU}}_{\mathrm{pred}}^{\mathrm{truth}}={\mathrm{Class}}_{i}|\mathrm{Object}\times \mathrm{Object}\times {\mathrm{IOU}}_{\mathrm{pred}}^{\mathrm{truth}} \end{array}$$

The final layers predict both the coordinates of their bounding boxes and their associated class probabilities. Then, the bounding boxes are normalized to fall between 0 and 1. All further layers increase non-linearity by using the leaky rectified linear activation function, as described in Equation (Equation 2), with an exception of the final layer, which employs a linear activation function:(2)$$f\left(x\right)=\left\{\begin{array}{cc} x, & \mathrm{if}\;x>0.\\ 0.1x, & \mathrm{otherwise}. \end{array}\right.$$

In 2016, YOLOv2 was released, improving on the original model by including batch normalization, anchor boxes, and dimension clusters [40]. Similarly, YOLOv3 was released in 2018, improving the model’s performance by employing a more efficient backbone network, incorporating a feature pyramid, and employing focal loss [41]. The initial model of YOLO had many successors, including YOLOv4, YOLOv5, YOLOv6, and YOLOv7. In 2023, YOLOv8 was released by Ultralytics [39]. Figure 5 shows the detailed architecture of YOLOv8. Figure 6, Figure 7, Figure 8 and Figure 9 show images of the dataset.

In the backbone architecture, the C2f module, based on Cross Stage Partial (CSP), is used in YOLOv8, as opposed to the C3 Module used in YOLOv5. The architecture of CSP enhances the learning capacity of CNN and decreases the computational effort of the model. The C2f module comprises two Conv Modules and n BottleNeck, connected through Split and Concat. The remainder of the backbone park is the same as that in YOLOv5. At the final layer of the backbone, the SPPF Module is used.

### 3.3. Movement Tracking

Once successfully able to detect vehicles in a parking lot, object tracking algorithms could be deployed to track the vehicles. Two tracking algorithms were implemented in the research. The first algorithm was DeepSORT. Figure 10 shows the workflow of our algorithm. The DeepSORT algorithm was implemented to track each vehicle throughout the frame. DeepSORT is an extension of SORT (Simple Online Realtime Tracking) [11]. DeepSORT uses appearance descriptors to minimize identity shifts, increasing the effectiveness of tracking. For problems involving the prediction of temporal or time series data, Kalman filtering is the used algorithm.

The second tracking algorithm that was implemented was Observation-Centric SORT (OC-SORT) [38]. OC-SORT was published in CVPR 2023. On numerous datasets, including MOT17, MOT20, KITTI, head tracking, and, particularly, DanceTrack, where the object motion is very non-linear, OC-SORT produces state-of-the-art results. To improve the accuracy and robustness during a period of occlusion, OC-SORT implements object observations to compute a virtual trajectory. Figure 11 shows the pipeline of OC-SORT.

### 3.4. Time Violation

After successful integration of vehicle detection, followed by vehicle tracking that assigns a unique ID to each unique vehicle, a simple time violation algorithm was deployed to track every car in the frame. Every minute, the presence of the vehicle was checked. If the vehicle did not exist in the next 10 consecutive frames, then the ID of the vehicle was discarded. On the other hand, if the ID was present in 15 consecutive frames, the vehicle was determined to have violated the time restriction. In the figure, a blue label is associated with no violation, while a red label is associated with violation.

### 3.5. Experimental Setup

This section discusses the experimental platform, providing details of the in-depth flow of the algorithm, including the chosen parameters. The workstation used for the experiment was Ubuntu Linux. All experiments were conducted in a 3.6 GHz Intel Xeon Quad-Core processor with 8GB RAM and an NVIDIA Quadro P4000 graphics card.

A Mean Average Precision (mAP) of 53.9 was achieved by YOLOv8x, which was higher than all other YOLO versions. Therefore, YOLOv8x was chosen for vehicle detection. A pretrained model was used, as the class was already trained on the MS COCO dataset. The resolution of the video feed was 1080P (1920 × 1080), with a frame rate of 15 fps. For YOLOv8, the image-size parameter was set to 640. The model thus resized the longest dimension to be 640, i.e., size 1920 became 640, while maintaining the aspect ratio. Therefore, the resized images were close to 640 × 360. Classes were filtered to be (2, 5, 7) i.e., car, bus, and truck, as the violation algorithm was checking for big vehicles. The confidence threshold was set to 0.5. The probability of the class occurring in the bounding box was evaluated using the confidence score. The maximum object detection parameter was set to 100.

For tracking vehicles, DeepSORT/OC-SORT tracking algorithms were used. The max-age parameter was set to 10. For DeepSORT algorithm, Max Age preserves the ID of a vehicle for a threshold number of frames before deleting the ID. The n_init parameter was set to 2. The n_init parameter refers to the number of objects detected before initializing the tracking of the object. max_cosine_distance was set to 0.3. Max cosine distance was the threshold to identify vehicle similarity by the DeepSORT algorithm. The trajectory parameter was set to False, as the trajectory of the vehicle over time was not required.

### 3.6. Validation Criteria

Object detection algorithms such as YOLOv8 use Mean Average Precision (mAP) to validate the performance of object detection. Since the objective of this work was to track vehicles across a frame, it was essential to validate the results with an object tracking algorithm. MOTA (Multiple Object Tracking Algorithm) metrics were used to validate the results. MOTA metrics measure the accuracy of both detection and tracking algorithms. Firstly, it was essential to define True Positive (TP), False Positive (FP), False Negative (FN), and True Negative (TN). Table 2 shows the detailed definitions. The formula for MOTA is shown in Equation (Equation 3). FN refers to no match for the ground truth object. This can occur because of various reasons, such as the vehicle being too far away from the camera, weather, different illumination, etc. FP refers to no ground truth object but a match by the algorithm. There was one instance of FP where the reflection of the vehicle was considered as a vehicle. False IDs refer to identification switches between the tracked vehicles. GT refers to total ground truth object.
(3)$$\begin{array}{c} MOTA=1-\frac{\sum_{t}FN_{t}+FP_{t}+IDS_{t}}{\sum_{t}GT_{t}} \end{array}$$

## 4. Results and Discussion

Figure 12 illustrates samples of vehicle detection and tracking, using YOLOv8 with two different object tracking algorithms across various daylight and weather settings.The performance of YOLOv8 with DeepSORT proved to be superior for the dataset in contrast to the recently introduced OC-SORT algorithm, which had exhibited notable advancements across numerous MOT datasets. MOTA scores of (1.0, 1.0, 0.96, 0.90) and (1, 0.76, 0.90, 0.8) were achieved for DeepSORT and OC-SORT tracking algorithms as shown in Table 3. The OC-SORT algorithm did not exhibit satisfactory results on this specific dataset. It consistently encountered challenges in maintaining consistent vehicle identification, particularly in the presence of rainy conditions. The algorithm frequently switched vehicle IDs, leading to inaccuracies in tracking which impacted the overall performance. In Figure 13 and Figure 14, the tracking of each algorithm on the Location 2 dataset can be observed. In the provided sample frame (f23401) from both Figure 13 and Figure 14, it is observed that the DeepSORT algorithm successfully tracked the vehicle IDs, while the OC-SORT algorithm struggled to accurately track the IDs. The challenges faced by OC-SORT in correctly tracking the IDs could potentially be attributed to factors such as rainy weather conditions and the presence of visually similar vehicles in close proximity. These factors may have led to difficulties in distinguishing between vehicles, resulting in ID switching. The robustness of DeepSORT in handling such scenarios could be attributed to its utilization of deep appearance feature embedding, which allows for more reliable tracking, even in challenging conditions. In the case of Location 1, the count for False Negatives (FN) was observed to be 3 and 4 for DeepSORT and OC-SORT, respectively, due to the presence of reflections from vehicles on the windows. These reflections interfered with the vehicle detection process, causing some vehicles to be falsely labeled as negatives which, thus, affected the MOTA scores.

Overall, based on the experiment conducted, it is clear that the performance of the vehicle tracking algorithm is greatly influenced by factors such as camera position and weather variation, particularly when using a single camera for inference. To achieve optimal tracking results, positioning the camera at or near a top-view perspective can greatly enhance the performance of the vehicle tracking algorithm. This camera placement minimizes the occurrence of vehicle overlapping, leading to improved tracking accuracy. By capturing a top-down view of the monitored area, the camera can provide a clear and unobstructed line of sight to the vehicles, reducing the chances of them overlapping or obstructing each other. This unambiguous view allows the tracking algorithm to more accurately detect and track individual vehicles, ensuring reliable and precise tracking results. Thus, selecting a top-view perspective for camera placement is crucial in minimizing vehicle overlap and maximizing the effectiveness of the vehicle tracking algorithms.

## 5. Conclusions

Manual parking time violation tracking is unideal, due to its requirements for manpower and high cost. In this research paper, a parking time violation tracking algorithm was successfully demonstrated; this has the potential to reduce additional manpower and labor costs. The algorithm can be implemented in many parking lot settings. The YOLOv8 algorithm was used to detect vehicles in the parking lot, while DeepSORT and OC-SORT were used to track the vehicles throughout the frame. For the DeepSORT algorithm, the MOTA scores were recorded as (1.0, 1.0, 0.96, 0.90) across four different surveillance datasets. This indicates a high level of accuracy and effectiveness in tracking multiple objects. On the other hand, the OC-SORT algorithm achieved MOTA scores of (1.0, 0.76, 0.90, 0.83) across the same datasets. While still achieving relatively high scores, there is a slight variation compared to DeepSORT, particularly in the second dataset, where the MOTA score dropped to 0.76 due to frequent ID switching. Overall, both algorithms showcased their ability to effectively track objects in the given surveillance data. These results provide valuable insights into the tracking performance of each algorithm and can aid in selecting the most suitable algorithm based on specific tracking requirements and dataset characteristics. The performance of the detection model might be increased by customized training of the model with vehicles. This is not necessary, but may help the algorithm to become more robust.

In future studies, researchers can explore the synchronization of the algorithm with two or more cameras. This synchronization would enable the algorithm to detect parking time violations in situations where multiple CCTV cameras are deployed. By integrating the feeds from multiple cameras, the algorithm can gather a more comprehensive view of the parking lot, improving the accuracy of violation detection. This scalability is important for expanding the algorithm’s applicability to larger parking areas or scenarios where multiple cameras are already in use. 

## Figures and Tables

**Figure 1 sensors-23-05843-f001:**
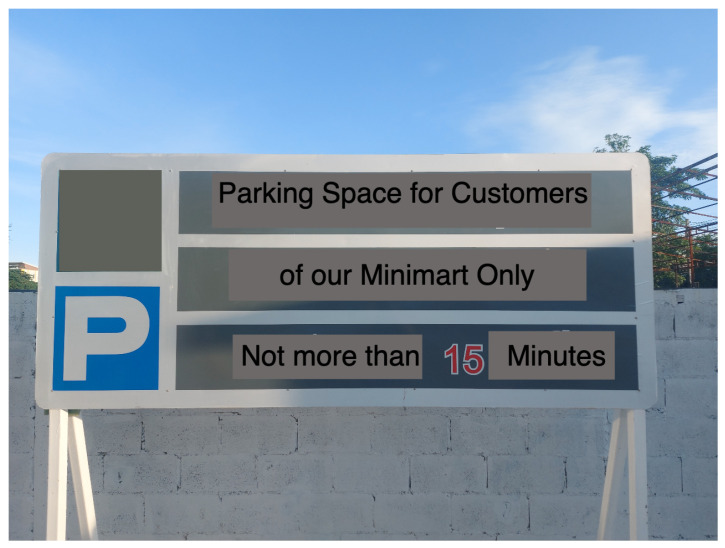
Representative parking restriction image, imposing a parking time limit of 15 min for consumers.

**Figure 2 sensors-23-05843-f002:**
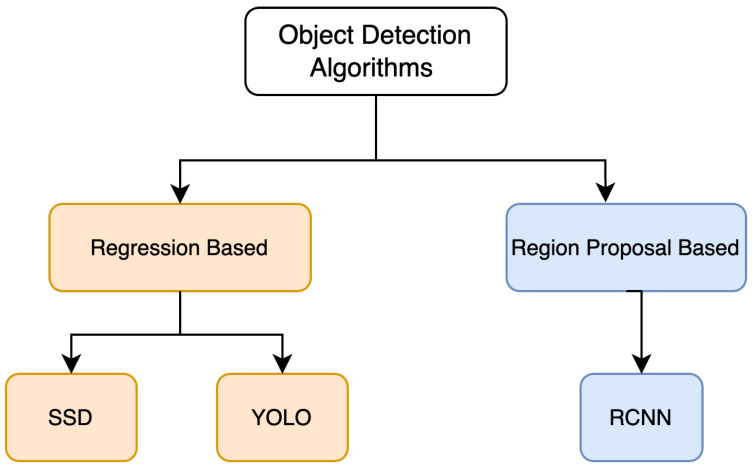
Object detection algorithms.

**Figure 3 sensors-23-05843-f003:**
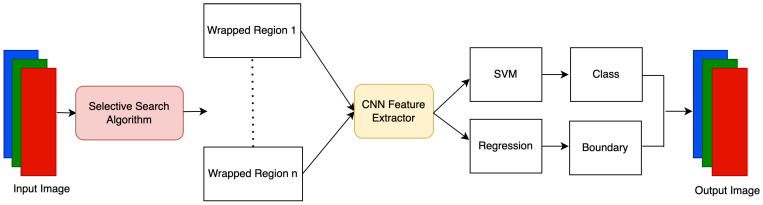
Flowchart of region-based CNN.

**Figure 4 sensors-23-05843-f004:**
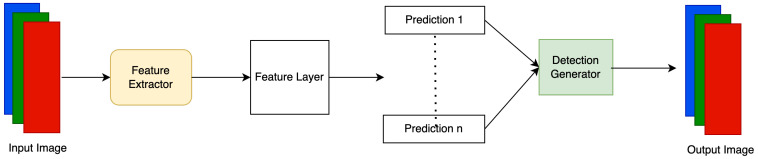
Flowchart of Single Shot MultiBox Detector.

**Figure 5 sensors-23-05843-f005:**
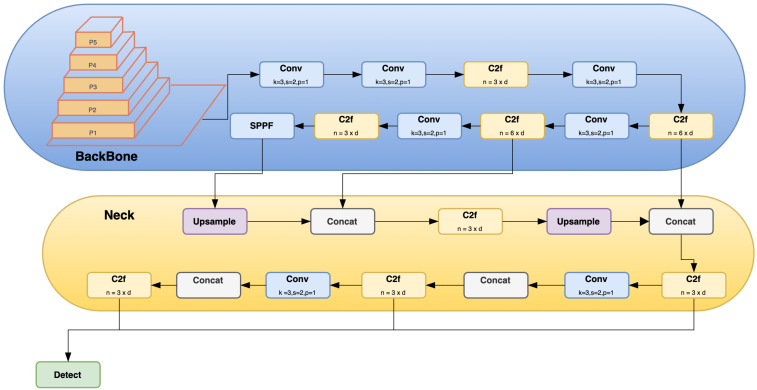
YOLOv8 architecture.

**Figure 6 sensors-23-05843-f006:**
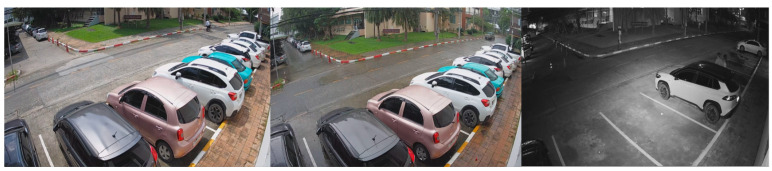
CCTV footage from Location 1.

**Figure 7 sensors-23-05843-f007:**
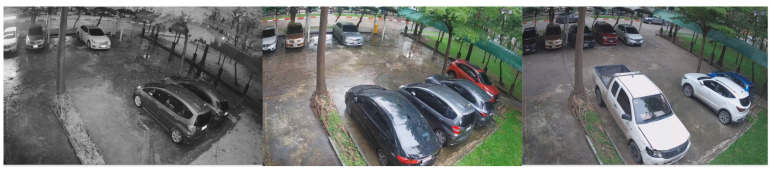
CCTV footage from Location 2.

**Figure 8 sensors-23-05843-f008:**
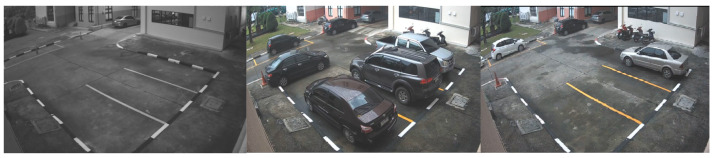
CCTV footage from Location 3.

**Figure 9 sensors-23-05843-f009:**
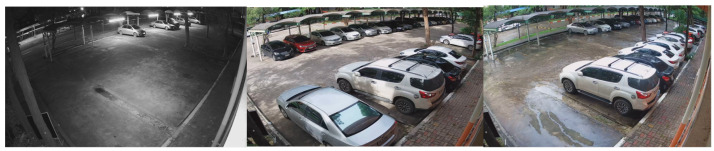
CCTV footage from Location 4.

**Figure 10 sensors-23-05843-f010:**
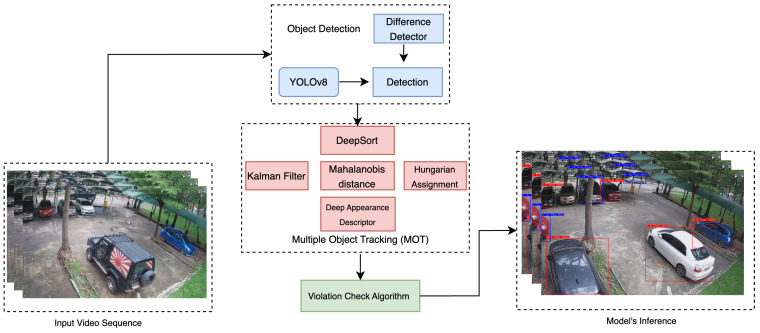
Workflow of our proposed work.

**Figure 11 sensors-23-05843-f011:**
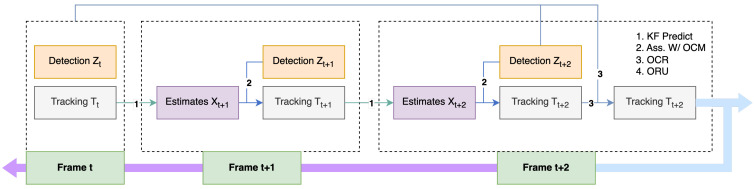
Pipeline of OC-SORT.

**Figure 12 sensors-23-05843-f012:**
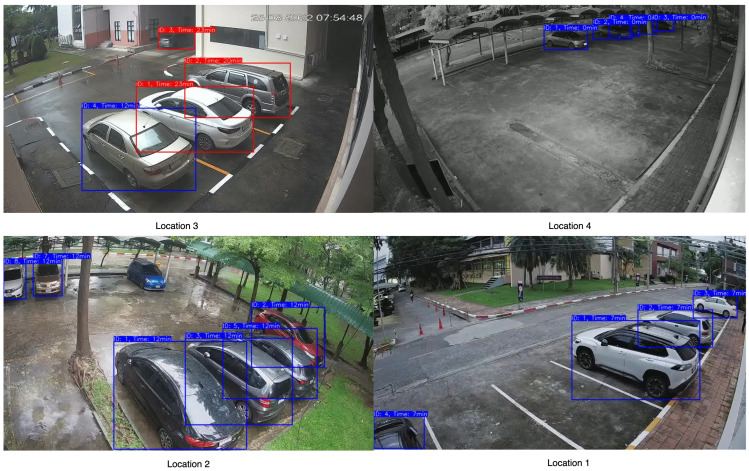
Sample images of detection and tracking by our model.

**Figure 13 sensors-23-05843-f013:**
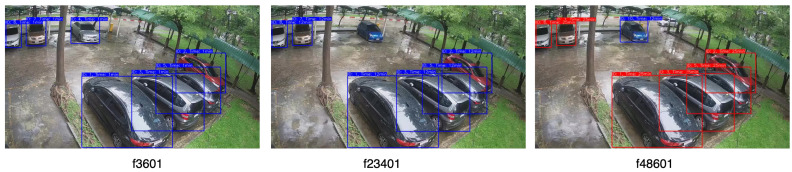
Tracking of DeepSORT on the Location 2 dataset.

**Figure 14 sensors-23-05843-f014:**
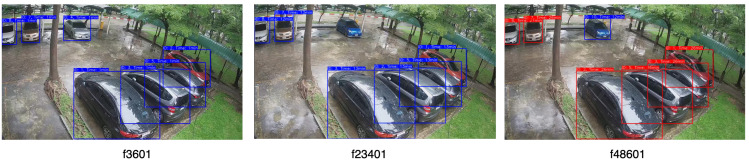
Tracking of OC-SORT on the Location 2 dataset.

**Table 1 sensors-23-05843-t001:** Results on the DanceTrack test set.

Tracker	HOTA	DetA	AssA	MOTA	IDF1
SORT [36]	47.9	72.0	31.2	91.8	50.8
DeepSORT [11]	45.6	71.0	29.7	87.8	47.9
ByteTrack [37]	47.3	71.6	31.4	89.5	52.5
OC-SORT [38]	54.6	80.4	40.2	89.6	54.6
OCSORT + Linear Interp [38]	55.1	80.4	40.4	92.2	54.9

**Table 2 sensors-23-05843-t002:** Definition and description of TP, TN, FP, and FN.

Definition	Description
True Positive (TP)	Vehicle is present and the algorithm can track vehicle.
True Negative (TN)	Vehicle is not present and the algorithm does not track vehicle
False Positive (FP)	Vehicle is not present but the algorithm tracks vehicle.
False Negative (FN)	Vehicle is present but the algorithm does not track vehicle

**Table 3 sensors-23-05843-t003:** Performance comparison of the two tracking algorithms.

Model	Dataset	FP	FN	IDS	MOTA
YOLOv8 + DeepSORT	Location 4	0	0	0	1
	Location 2	0	0	0	1
	Location 3	1	0	0	0.96
	Location 1	0	3	0	0.90
YOLOv8 + OC-SORT	Location 4	0	0	0	1
	Location 2	0	0	7	0.76
	Location 3	1	0	2	0.90
	Location 1	0	4	1	0.83

## Data Availability

The data presented in the research paper are available on request, as they represent private parking lot data from KMITL.

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
