# Peer review of "Parking Time Violation Tracking Using YOLOv8 and Tracking Algorithms"

_sensors, 2023, doi:10.3390/s23135843_

Round 1

Reviewer 1 Report

The paper presented parking time violations using YOLOv8 and Deepsort tracking algorithm. The area is of high interest and well presented. However, there are some major points that need to be addressed before publication.

1) The related work should be improved by adding more detection and tracking algorithm. There should be a summarized table of the tracking algorithms that highlighted their features.

2) The dataset details are missing. There should be a subsection about the dataset (with its limitations)

3) Results are not described in detail. Also, there should be more Quantitative analysis of the results that can show the efficiency/accuracy of the proposed approach.

4) A comparative analysis is missing. There should be a comparative analysis of more than one tracking algorithm. (Apply some more state-of-the-art tracking algorithms and compare the results).

5) There should be a discussion section that describes the effectiveness of the proposed methodology wrt the existing solution. 

6) Follow some standard references style.

The paper should be proofread again to remove grammatical mistakes. 

Reviewer 2 Report

The purpose of this paper is to study the detection of parking time violations and explore low-cost solutions to solve the problem. The structure of the article is well-organized and clear in its hierarchy, with relevant literature referenced closely linked to the theme. The paper proposes a new algorithm for detecting parking time violations that utilizes Yolov8 and DeepSORT tracking algorithms to track vehicles across consecutive frames. The paper is written in a standardized manner, and its structure is logical. However, the language in the introduction section is overly complex and lacks conciseness. There are specific issues in the article, including:

1.  The YOLOv8+DeepSORT detection approach used in this paper has already been researched for pedestrian and vehicle detection, so the innovation of this paper is insufficient.

2.  Through the introduction section, it is evident that there are sufficient application scenarios for the research direction of this paper in the author's region. However, in other developed countries, the detection systems for illegal parking have already been well developed, and there is a lack of application scenarios.

3.  The experiment part of the article lacks any comparative experiments, and the overall experimental structure does not conform to the academic writing norms of a research paper.

4. The paper mentions that it proposes a time-based method for detecting illegal parking for the first time, but it still uses the Kalman filter algorithm for state estimation, an outdated algorithm.

Round 2

Reviewer 1 Report

The comments are addressed well. 

Need a proofread before the final version